# Effect of Korean Red Ginseng and Rg3 on Asian Sand Dust-Induced MUC5AC, MUC5B, and MUC8 Expression in Bronchial Epithelial Cells

**DOI:** 10.3390/molecules26072002

**Published:** 2021-04-01

**Authors:** Seung-Heon Shin, Mi-Kyung Ye, Dong-Won Lee, Byung-Jun Kang, Mi-Hyun Chae

**Affiliations:** Department of Otolaryngology-Head and Neck Surgery, School of Medicine, Catholic University of Daegu, Daegu 42472, Korea; miky@cu.ac.kr (M.-K.Y.); neck@cu.ac.kr (D.-W.L.); rkdqudwns03@naver.com (B.-J.K.); leonen@hanmail.net (M.-H.C.)

**Keywords:** Korean red ginseng, ginsenoside Rg3, bronchial epithelial cell, mucin, transcription factor

## Abstract

Korean Red ginseng (KRG), commonly used in traditional medicine, has anti-inflammatory, anti- oxidative, and anti-tumorigenic properties. Asian sand dust (ASD) is known to aggravate upper and lower airway inflammatory responses. BEAS-2B cells were exposed to ASD with or without KRG or ginsenoside Rg3. Mucin 5AC (MUC5AC), MUC5B, and MUC8 mRNA and protein expression levels were determined using quantitative RT-PCR and enzyme-linked immunosorbent assay. Nuclear factor kappa B (NF-κB), activator protein 1, and mitogen-activated protein kinase expression and activity were determined using western blot analysis. ASD induced MUC5AC, MUC5B, and MUC8 mRNA and protein expression in BEAS-2B cells, which was significantly inhibited by KRG and Rg3. Although ASD-induced mucin expression was associated with NF-κB and p38 mitogen-activated protein kinase (MAPK) activity, KRG and Rg3 significantly suppressed only ASD-induced NF-κB expression and activity. KRG and Rg3 inhibited ASD-induced mucin gene expression and protein production from bronchial epithelial cells. These results suggest that KRG and Rg3 have potential for treating mucus-producing airway inflammatory diseases.

## 1. Introduction

Asian sand dust (ASD), originating from sandstorms in the Gobi Desert and the Ocher Plateau, is inhaled and comes into contact with respiratory epithelial cells, thereby inducing neutrophilic or eosinophilic lung inflammation by stimulating the production of inflammatory mediators [1,2]. The organic and inorganic compounds of ASD, such as SiO_2_, Al_2_O_3_, Fe_2_O_3_, et al., influence the development of upper and lower airway inflammation, and increased concentration of ASD in the air correlates with asthma severity and adverse respiratory health effects [3]. Particulate matter less than 10 μm (PM10)-main components of ASD- is associated with pulmonary dysfunction, cardiovascular disease, and hepatic fibrogenesis [4,5]. Daily mortality and hospital admission rate increase due to the deterioration of respiratory function during the ASD season [3,6]. ASD influences morbidity and mortality in inflammatory airway diseases by increasing mucin gene expression in upper and lower airway epithelial cells, which causes mucus production, aggravation of respiratory symptoms and severity [7,8].

Mucus overproduction and hypersecretion are frequently observed in several airway diseases. Mucins comprise about 2% of mucus with MUC2, MUC4, MUC5AC, MUC5B, and MUC8 commonly found in airway mucosa. MUC5AC and MUC5B constitute an important component of secretory mucin in airway diseases and are the subjects of frequent study [9,10]. MUC8 mRNA and protein levels were increased in sinus mucosa in chronic rhinosinusitis and in lung of cystic fibrosis, but the physiological functions of MUC8 remain unclear [11,12].

Heat-processing Panax ginseng Meyer converts ginsenoside compounds within the root to yield Korean Red ginseng (KRG). Since red ginseng has higher biological effects and fewer side effects than fresh ginseng, KRG is commonly used in traditional medicine throughout East Asia for its immunomodulatory, anti-allergic, anti-inflammatory, anti-oxidative, and anti-tumorigenic properties [13,14,15]. Various ginsenosides regulate inflammatory reactions through the regulation of cytokine, chemokine, and cyclooxygenase-2 production [16,17]. KRG is composed of more than 40 ginsenosides, although not all of them significantly affect airway inflammatory responses [17]. Traditional usage suggests that KRG or ginsenosides have therapeutic potential for treating airway inflammatory diseases. Ginsenoside Rg3 (Rg3), a member of protopanaxadiols, is the main component of KRG with anti-inflammatory and anti-cancer properties. Anti-inflammatory effects of Rg3 suppress nitric oxide, reactive oxygen species, prostaglandin E2 and proinflammatory cytokine production [18]. This anti-cancer effect is associated with induction of apoptosis, induction of autophagy, and inhibition of angiogenesis [19]. However, the effects of KRG and Rg3 on mucin production in airway epithelial cells are not commonly studied. In this study, we investigated the effects of KRG and ginsenoside Rg3 on ASD-induced mucin production in bronchial epithelial cells.

## 2. Results

### 2.1. The Effects of ASD, KRG, and Rg3 on Cell Viability

To determine the optimal dose of ASD, KRG, and Rg3 and optimal treatment time, we performed a cell proliferation assay. BEAS-2B cells were treated with various concentrations of ASD, KRG, and Rg3 for 24, 48, and 72 h. Cell viability was significantly decreased at 500 and 1000 μg/mL of Rg3 (Figure 1B) and 250 and 500 μg/mL of ASD (Figure 1C) over 24 h incubation, but concentrations less than 500 μg/mL of KRG did not affect the survival (Figure 1A) of BEAS-2B cells. When treated with 50 μg/mL of ASD and 500 μg/mL of KRG or 50 μg/mL of Rg3, bronchial epithelial cells’ survival was not significantly affected over 72 h (Figure 1D). Therefore, we used 50 and 100 μg/mL of ASD, 500 μg/mL of KRG, and 50 μg/mL of Rg3 for further experiments. The IC50 values of ASD, KRG, and Rg3 were derived from dose-response curves (24 h: KRG: 995 μg/mL, Rg3 210 μg/mL, and ASD 282 μg/mL; 48 h: KRG 1130 μg/mL, Rg3 148 μg/mL, and ASD 186 μg/mL; 72 h: KRG 613 μg/mL, Rg3 139 μg/mL, and ASD 172 μg/mL, respectively).

### 2.2. The Effects of KRG and Rg3 on ASD Induced MUC5AC, MUC5B, and MUC8 mRNA Expression

When BEAS-2B cells were treated with 100 μg/mL of ASD, MUC5AC, MUC5B, and MUC8 mRNA expression was significantly increased compared to unstimulated cells. In contrast, 50 μg/mL of ASD had no effect on mRNA expression. When the BEAS-2B cells were pretreated with 500 μg/mL of KRG or 50 μg/mL of Rg3, ASD-induced MUC5AC and MUC5B mRNA expression was significantly inhibited (MUC5AC; KRG, 45.9 ± 13.7%, Rg3 42.1 ± 21.4% and MUC5B; KRG 65.6 ± 17.4%, Rg3 56.2 ± 23.2%, respectively). However, KRG and Rg3 did not influence ASD induced MUC8 mRNA expression (Figure 2).

### 2.3. The Effects of KRG and Rg3 on ASD Induced MUC5AC, MUC5B, and MUC8 Protein Expression

As with mRNA levels, cells treated with 100 μg/mL of ASD expressed significantly higher levels of MUC5AC, MUC5B, and MUC8 protein than unstimulated cells. 50 μg/mL of ASD again had no effect. When the cells were pretreated with 500 μg/mL of KRG or 50 μg/mL of Rg3, ASD-induced MUC5AC and MUC5B protein levels were significantly decreased (MUC5AC; KRG 25.0 ± 7.3%, Rg3 20.2 ± 6.4%, MUC5B; KRG 22.2 ± 6.7%, Rg3 24.0 ±13.5%, and MUC8; KRG: 23.8 ± 12.1%, Rg3 29.5 ± 9.2%, respectively) (Figure 3).

### 2.4. The Effects of KRG and Rg3 on ASD-Induced Transcription Factors Expression

Using western blot analysis, we determined the effect of ASD on NF-κB, AP-1, and MAPK transcription factor expression. We found that while ASD enhanced NF-κB and phosphorylated-NF-κB expressions (Figure 4B), it did not affect AP-1 (Figure 4C) and MAPK (Figure 4D) expression. When cells were pretreated with KRG and RG3, ASD-induced increases in NF-κB and phosphorylated-NF-κB expressions were suppressed (Figure 4B). 

When BEAS-2B cells were pretreated with transcription factor inhibitors, MUC5AC, MUC5B, and MUC8 protein production was significantly and selectively inhibited by NF-κB and p38 MAPK inhibitors (Figure 5).

## 3. Discussion

ASD contains various sizes (0.1–20 μm in diameter) of particles and PM10, fine, and ultrafine particles may influence respiratory inflammatory diseases, such as bronchial asthma, rhinosinusitis, and allergic rhinitis. Mucus hypersecretion is a major pathognomonic finding in both upper and lower airway inflammation. Among the mucin genes, MUC5AC, MUC5B, and MUC8 are the major secretory mucin genes implicated in inflammatory airway diseases [20]. In this study, we found that ASD treatment induced increases in MUC5AC, MUC5B, and MUC8 mRNA and protein expression in bronchial epithelial cells and that these increases were associated with the NF-κB and p38 MAPK pathway. We tested whether KRG and the ginsenoside Rg3 could suppress ASD-induced mucin gene and protein expression in bronchial epithelial cells, finding that they did so through the downregulation of NF-κB.

KRG is known to possess various biological and immunological activities including anti-inflammatory, anti-oxidative, and anti-tumorigenic properties [13,14,15]. Ginsenosides, the primary active constituents of ginseng, are saponins with steroid-like hydrophobic backbones connected to sugar moieties. Treating fresh ginseng with heat converts it into red ginseng and increases the concentrations of ginsenosides Rg2, Rg3, and Rh1 [21]. KRG and various pure ginsenosides inhibited lung inflammatory responses through the inhibition of MAPK, NF-κB, and c-Fos activation [17]. Although, not all of the ginsenosides isolated from KRG influenced inflammatory responses, Rg3 has an anti-inflammatory effect via the reduction of COX-2, inducible nitric oxide synthase, and proinflammatory cytokines [22,23].Ginsenoside Rb1 was found to inhibit lipopolysaccharide-induced MUC5AC expression in human airway epithelial cells [24]. We evaluated the optimal concentration for the inhibition of mucin gene expression by pretreatment of BEAS-2B cells with various concentrations of KRG (50 to 500 μg/mL) and Rg3 (5 to 50 μg/mL). KRG and RG3 inhibited MUC5AC, MUC5B and MUC8 mRNA expression in a dose dependent manner (data not shown), with 500 μg/mL of KRG and 50 μg/mL of Rg3 used as an optimal dose in this study. ASD-induced MUC MUC5 AC, MUC5B, and MUC8 mRNA and protein expression through NF-κB and p38 MAPK signaling pathways. Inhibition of the ERK, JNK, and AP-1 transcription factors did not influence the production of mucin proteins in these cells. Choi et al. previously reported that ASD-induced MUC5B and MUC8 expression via TLR4-dependent ERK2 and p38 MAPK pathways in NCI-H292 cells and primary nasal epithelial cells [8]. Moreover, ASD-induced the production of ROS and proinflammatory cytokines via the MAPK signaling pathway in nasal fibroblasts [25]. Lipopolysaccharide induced mucus hypersecretion was associated with TLR4 and NF-κB signaling pathway in bronchial epithelial cells [26]. It is thus apparent that ASD leads to the production of chemical mediators or mucus hypersecretion using different pathophysiologic mechanisms dependent on the type of cells studied. Various transcription factors such as NF-κB, AP-1, and ERK2 are responsible for regulation of mucin gene and protein expression in different airway epithelial cells in response to various stimuli [8,24,26]. NF-κB sites on MUC promoters in particular perform a crucial function in regulating MUC expression in bronchial epithelial cells [22]. KRG and ginsenosides show a wide range of anti-inflammatory action, and the mechanisms of action include inhibition of kinase phosphorylation, NF-κB induction, NF-κB translocation, and chemical mediator production [16]. Rg3 exerts an anti-inflammatory effect through the attenuation of the NF-κB signaling pathways in airway epithelial cell and asthmatic airway tissues [22]. This study demonstrates that KRG and Rg3 inhibit ASD-induced MUC5AC, MUC5B, and MUC8 mRNA and protein expression through the inhibition of NF-κB expression and activation independent of P38 MAPK.

KRG contains approximately 40 types of ginsenosides, and other active pharmaceutical constituents. Some ginsenosides are transformed into active metabolites by gut microbiota and are absorbed into the blood before they can exert pharmacological effects, while others have no effect on inflammatory responses of airway epithelial cells [17,27]. Upper and lower airway inflammatory diseases are commonly treated with topical spray or inhalation agents, which make direct contact with airway mucosa to exert pharmacological action. Although we only studied Rg3, the data suggest that it could be a good candidate ginsenoside as a topical treatment to control mucus producing airway diseases.

## 4. Materials and Methods

### 4.1. Preparation of KRG and ASD

The standardized water extract of KRG and Rg3 were supplied by the KT&G Corporation (Daejeon, Korea). Panax ginseng Meyer was cultivated for six years in the Korean peninsula. Ginseng roots were collected and dried. KRG was manufactured by steaming and drying white ginseng, and the hot water extract was prepared and provided by KT&G. Ginsenoside Rg3 content was determined by high-performance liquid chromatography. Chemical structure of Rg3 is proposed in Figure 6.

ASD was collected from air dust using a high-volume air sampler (HV-500F, Sibata, Japan), during an ASD warning period in Incheon. After the dust was collected, the filter paper was washed with 10 mL of phosphate buffer solution (PBS). The fluid was filtered and the particulate matter were collected and then centrifuged. The collected ASD material was placed in a 1.5-mL tube and sterilized at 121 °C for 15 min. The sterilized ASD was stored in a −20 °C freezer.

### 4.2. Bronchial Epithelial Cell Culture and Cytotoxic Effect of ASD

Human bronchial epithelial BEAS-2B cells, transformed by adenovirus, were purchased from American Type Culture Collection (Rockville, MD, USA). Epithelial cells were cultured with DMEM/F12 medium supplemented with 100 international units of penicillin, 100 μg/mL streptomycin, 2 μg/mL of amphotericin B, and heat-inactivated 10% fetal bovine serum (Invitrogen, Carlsbad, CA, USA) at 37 °C and 5% CO_2_. Cell suspensions at 5 × 104 cells/well were grown to 80% confluence for further studies. 

To determine the cytotoxic effects of ASD, BEAS-2B cells were incubated with 0, 10, 100, or 500 μg/mL of ASD for 72 h. Cell cytotoxicity was determined using a CellTiter-96^®^ aqueous one solution cell proliferation assay kit (Promega, Madison, WI, USA). For this assay, tetrazolium compound and phenazine etho-sulfate were added to each well and incubated for 4 h at 37 °C in a 5% CO_2_ chamber. Color intensities were assessed using a microplate reader at wavelength of 490 nm. The cytotoxic effects of KRG (0 to 500 μg/mL) and Rg3 (0 to 1000 μg/mL) were also determined using a CellTiter-96^®^ aqueous one solution cell proliferation assay kit (Promega).

### 4.3. The Effect of KRG and Rg3 on ASD-Induced MUC5AC, MUC5B, and MUC8 mRNA Expression

MUC5AC, MUC5B, and MUC8 mRNA expression was measured after stimulation with 50 or 100 μg/mL of ASD for 48 h. To determine the effects of KRG and Rg3 on mucin gene expression, BEAS-2B cells were pretreated with 500 μg/mL of KRG or 50 μg/mL of Rg3 for 1 h. The cells were treated with 1 mL of TRIzole reagent (Roche Diagnostics, Mannheim, Germany), and RNA was extracted. Then RNA was treated with DNase to remove any contamination of DNA. RNA purity and concentration were measured using a spectrophotometer (Beckman, Mountain View, CA, USA). From amplified cDNA, quantitative polymerase chain reaction (PCR) of MUC5AC, MUC5B, MUC8 and β-actin in the same 96 well plate using a SYBR green PCR core kit (PE Applied Biosystems, Foster, CA, USA) was performed with the GeneAmp 5700 system (PE Applied Biosystems). The primer sequences and amplified products were as follows: MUC5AC sense 5′-TCA TCA TCC AGC AGG GCT-3′ and antisense 5′-CCG AGC TCA GAG GAC ATA TGG G-3′ (103 bp), MUC5B sense 5′-TGC CCC TTG TTC TGT GAC TT-3′ and antisense 5′-ACG CAC TTC ATC TGG TCC TC-3′ (194 bp), MUC8 sense 5′-GAC AGG GTT TCT CCT CAT TG-3′ and antisense 5′-CGT TTA TTC CAG CAC TGT TC-3′ (240 bp), and β-actin sense 5′-ACA GGA AGT CCC TTG CCA TC-3′ and antisense 5′-AGG GAG ACC AAA AGC CTT CA-3′ (248 bp). The annealing temperature was 54 °C for MUC5B, 60 °C for MUC5AC, and 56 °C for MUC8.

All samples were amplified in triplicate. mRNA expression levels were normalized to the median value of the endogenous control β-actin. The relative quantitation of mRNA levels was determined using the relative quantification 2-delta delta CT method.

### 4.4. The Effect of KRG and Rg3 on ASD Induced MUC5AC, MUC5B, and MUC8 Protein Production

MUC5AC, MUC5B, and MUC8 protein levels were determined using an enzyme linked immunosorbent assay (ELISA). BEAS-2B cell lysates were prepared at multiple dilutions and incubated at 40 °C until dry. Plates were washed with PBS and blocked with 2% bovine serum albumin. Thereafter, plates were incubated with 1:200 diluted MUC5AC, MUC5B, and MUC8 primary antibodies (Santa Cruz Biotechnology, Santa Cruz, CA, USA) in PBS containing 0.05% Tween 20 for 1 h. The wells were washed with PBS, and then incubated horseradish peroxidase-conjugated secondary antibodies for 4 h. Color was developed with a 3, 3′, 5, 5′-tetramethylbenzidine peroxidase solution and stopped with 2N H2SO4. Optical densities were measured for absorbance at 450 nm. Results were expressed as the ratio of baseline controls.

### 4.5. The Effect of KRG and Rg3 on ASD Induced Transcription Factor Expression 

After 1 h exposure with ASD and KRG or Rg3, BEAS-2B cells were harvested and lysed in an ice-cold lysis buffer (Thermo Scientific, Rockford, IL USA). Collected whole cell lysates were subjected to sodium dodecyl sulfate polyacrylamide gel electrophoresis to separate protein and transferred onto a nitrocellulose membrane (Bio-Rad, Berkeley, CA, USA). The membranes were blocked with 5% skim milk solution and incubated with the following antibodies against nuclear factor kappa B (NF-κB): phosphorylated NF-κB, C-Jun, phosphorylated C-Jun, p38, phosphorylated p38, ERK, phosphorylated ERK, JNK, phosphorylated JNK, and GAPDH (Santa Cruz Biotechnology). After 1 h incubation, membranes were washed with Tris-buffered saline with 0.1% Tween 20 and then treated with peroxidase-conjugated anti-rabbit immunoglobulin G (Santa Cruz Biotechnology). Bands were visualized using horseradish peroxidase conjugated secondary antibodies and an enhanced chemiluminescence system (Pierce, Rockford, IL, USA). Band densities were measured using the multi Gauge v.2.02 software (Fujifilm, Tokyo, Japan) and expressed as a percentage of treated versus untreated cells.

### 4.6. Effects of Transcription Factor Inhibitors on MUC5AC, MUC5B, and MUC8 Protein Expression

BEAS-2B cells were pretreated with the NF-κB inhibitor BAY 11-7082, the activator protein-1 (AP-1) inhibitor curcumin, the p38 inhibitor SB203580, the ERK inhibitor PD98059, and the JNK inhibitor SP600125 to interrogate the MAPK pathway (Calbiochem, San Diego, CA, USA). After a 1-h treatment, cells were stimulated with ASD for 48 h and then MUC5AC, MUC5B, and MUC8 protein expression determined using an ELISA method.

### 4.7. Statistical Analysis

All experiments were performed in triplicate and repeated at least five times with comparable results. Results are presented as mean ± standard deviation. Statistical significance to determine the cytotoxic effects of ASD and KRG or Rg3 was determined using single-factor repeated measure analysis. Student’s *t*-test was used for comparisons between two groups, while data comparisons between several groups were made using one-way analysis of variance (ANOVA) followed by Turkey’s test (SPSS ver. 21.0; IBM Corp., Armonk, NY, USA). A *p*-value of 0.05 or less was considered statistically significant.

## 5. Conclusions

This study demonstrated that KRG and Rg3 suppressed ASD-induced MUC5AC, MUC5B, and MUC8 mRNA and protein expression in BEAS2B bronchial epithelial cells through the inhibition of NF-κB. These results provide basic mechanistic information about the inhibition of mucus production by KRG and Rg3 associated with ASD-induced mucus hypersecretion. These results suggest that KRG and Rg3 could be used as a treatment strategy in patients with ASD-related mucus producing airway inflammatory diseases.

## Figures and Tables

**Figure 1 molecules-26-02002-f001:**
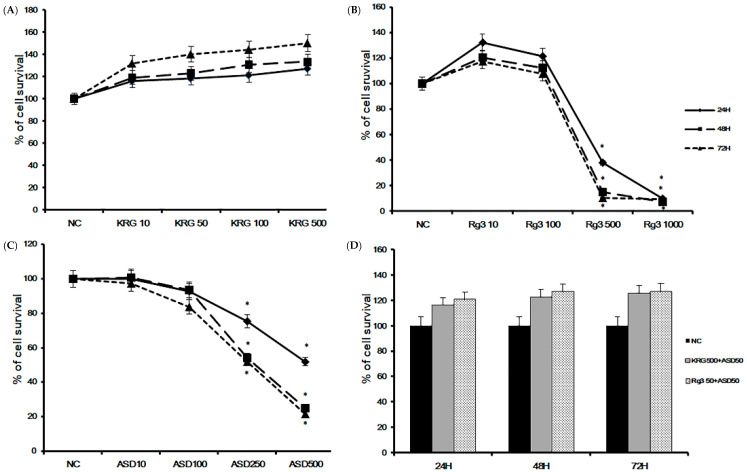
Cell viability of Korean red ginseng (KRG), Rg3, and Asian sand dust (ASD) on bronchial epithelial cells (BEAS-2B) at various concentrations and times using the CellTiter-96^®^ aqueous cell proliferation assay. Cell survival was significantly decreased at 500 μg/mL Rg3 (**B**) and 250 μg/mL ASD (**C**). However, less than 500 μg/mL of KRG (**A**) or 50 μg/mL of ASD and 500 μg/mL of KRG or 50 μg/mL of Rg3 (**D**) did not affect the survival of BEAS-2B cells. *: *p* < 0.05 compared with negative control (NC), *n* = 5.

**Figure 2 molecules-26-02002-f002:**
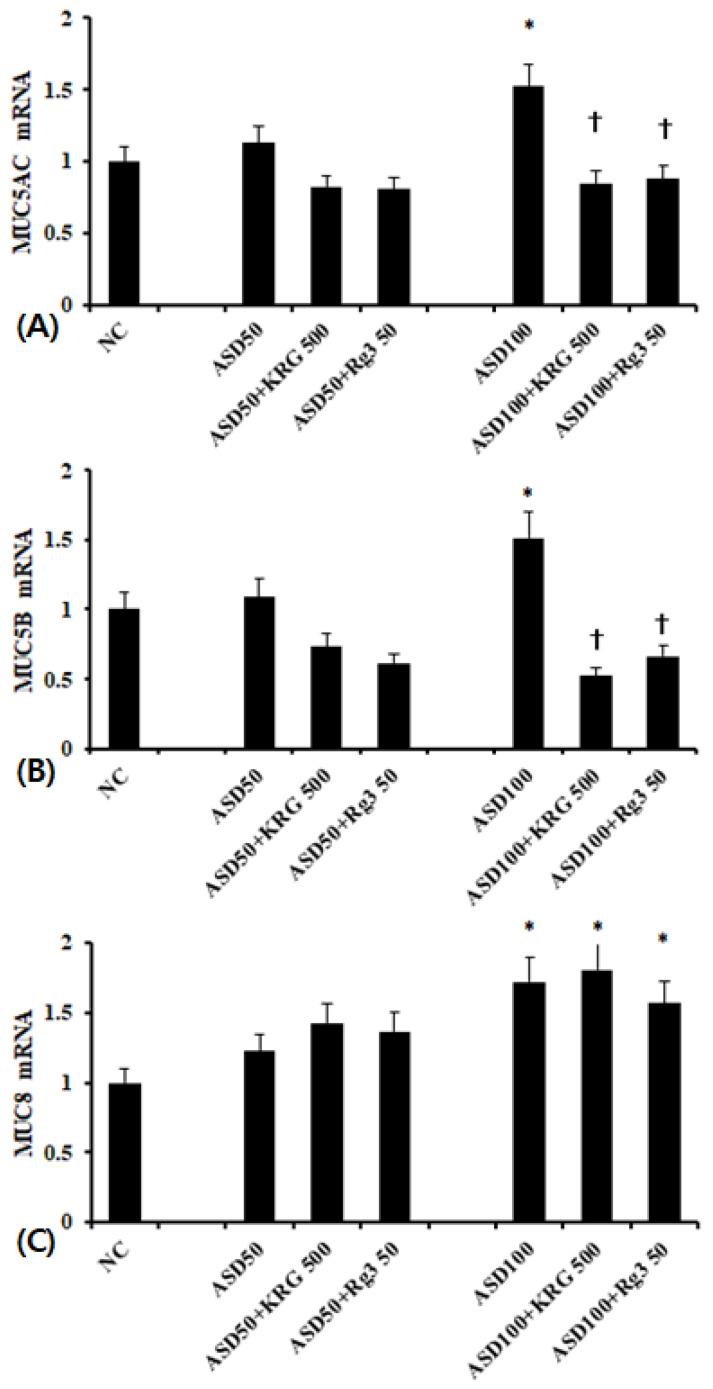
Effects of Korean red ginseng (KRG) and Rg3 on mucin gene expression in bronchial epithelial cells (BEAS-2B). 100 μg/mL Asian sand dust (ASD) significantly enhanced MUC5AC (**A**), MUC5B (**B**), and MUC5B (**C**) mRNA expression in BEAS-2B cells, which was suppressed by Korean red ginseng (KRG) and Rg3. NC; negative control, *; *p* < 0.05 compared with NC, †; *p* < 0.05 compared with ASD stimulated group, *n* = 7.

**Figure 3 molecules-26-02002-f003:**
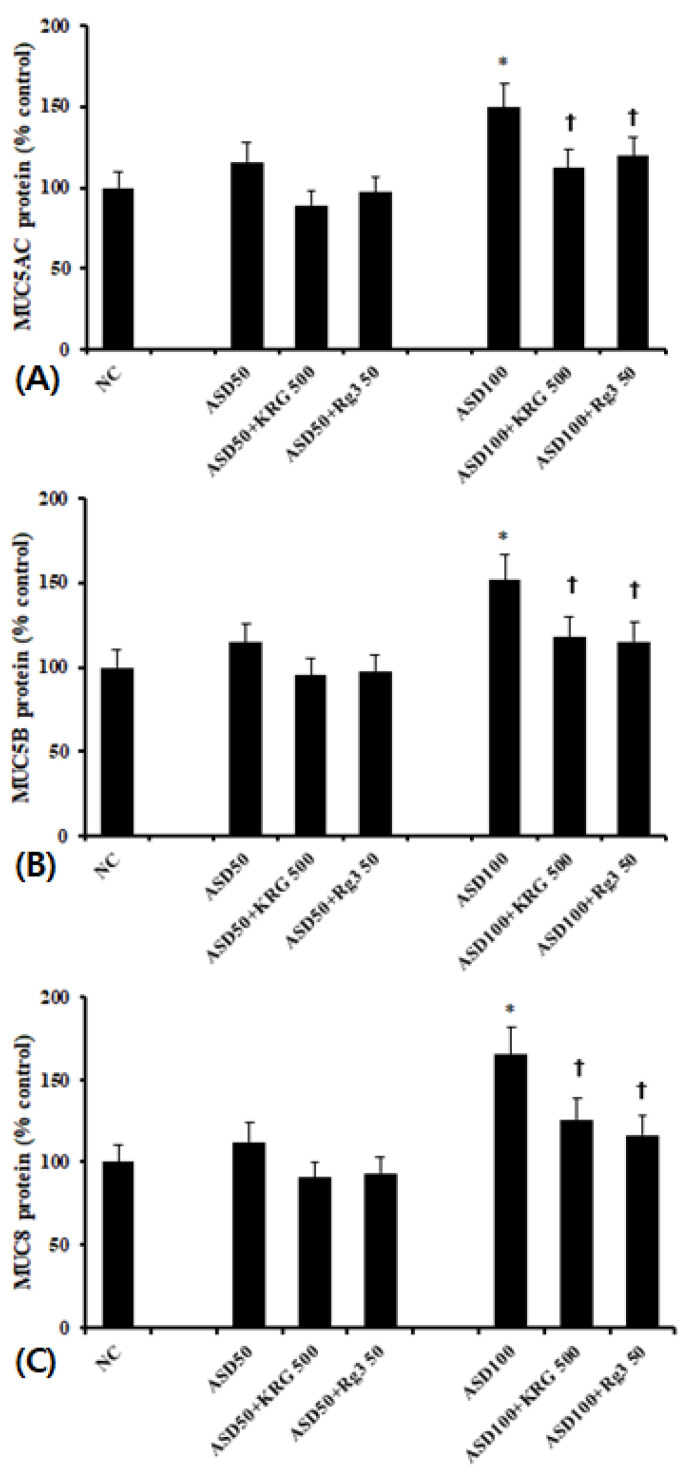
Effects of Korean red ginseng (KRG) and Rg3 on mucin protein expression in bronchial epithelial cells (BEAS-2B). 100 μg/mL Asian sand dust (ASD) significantly enhanced MUC5AC (**A**), MUC5B (**B**), and MUC8 (**C**) protein expression in BEAS-2B cells, which was suppressed by KRG and Rg3. NC; negative control, *; *p* < 0.05 compared with NC, †; *p* < 0.05 compared with ASD stimulated group, *n* = 5.

**Figure 4 molecules-26-02002-f004:**
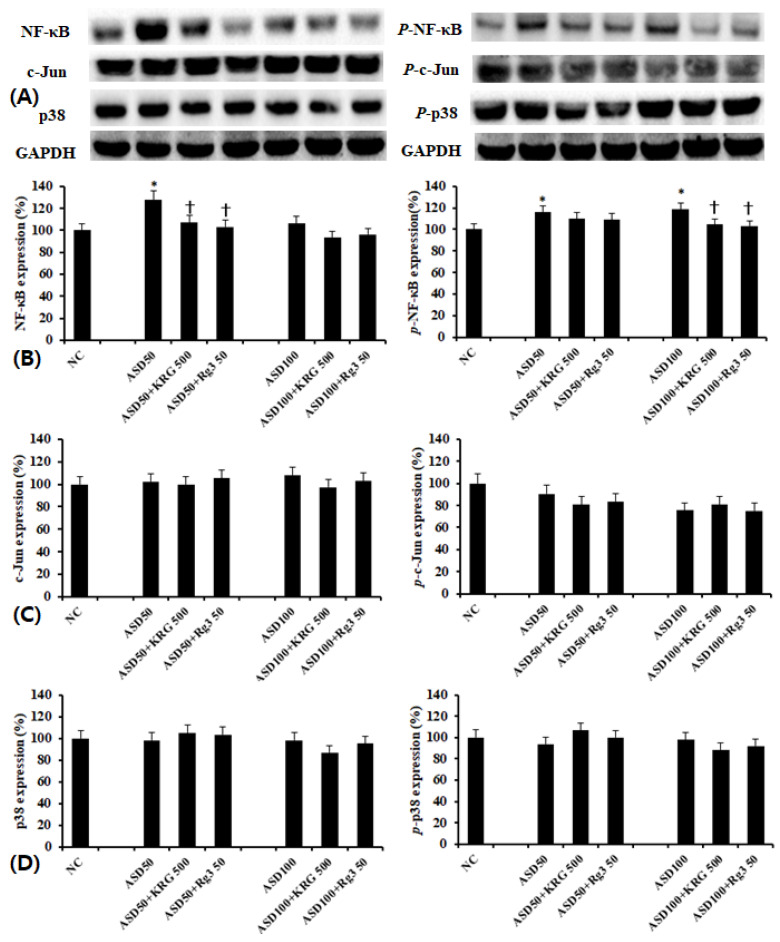
Effects of Korean red ginseng (KRG) and Rg3 on the transcription factor expression in bronchial epithelial cells (BEAS-2B). (**A**) shows representative results of western blot analysis of transcription factors. Asian sand dust (ASD) induced NF-κB and phosphorylated-NF-κB expression, which was significantly inhibited by KRG and Rg3 (**B**). However, KRG and Rg3 did not affect c-Jun (**C**) and p38 (**D**) expressions. NC; negative control, *; *p* < 0.05 compared with NC, †; *p* < 0.05 compared with ASD stimulated group, *n* = 5.

**Figure 5 molecules-26-02002-f005:**
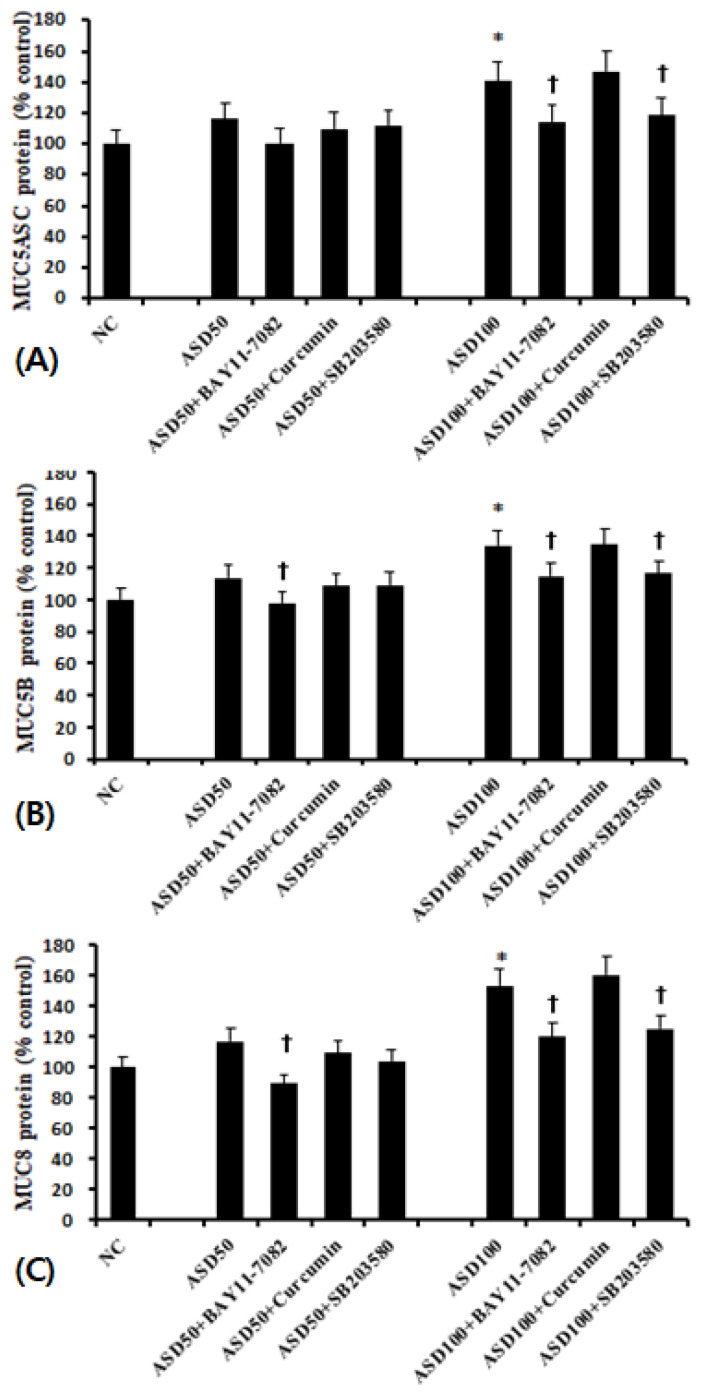
Effects of transcription factor inhibitors on the expression of mucin protein in bronchial epithelial cells (BEAS-2B). ASD-induced MUC5AC (**A**), MUC5B (**B**), and MUC8 (**C**) protein production was significantly inhibited by NF-κB (BAY 11-7082) and p38 MAPK (SB203580) inhibitors. *; *p* < 0.05 compared with NC, †; *p* < 0.05 compared with ASD stimulated group, *n* = 5.

**Figure 6 molecules-26-02002-f006:**
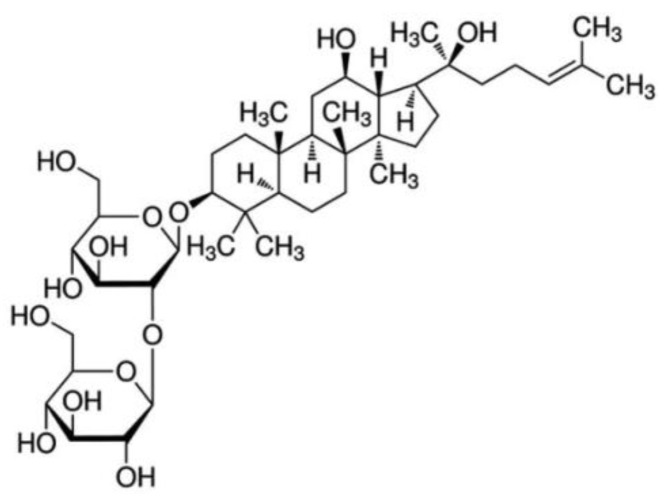
Structure of Rg3.

## Data Availability

The data presented in this study are available on request from the corresponding author.

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
