# Peer review of "Effect of Korean Red Ginseng and Rg3 on Asian Sand Dust-Induced MUC5AC, MUC5B, and MUC8 Expression in Bronchial Epithelial Cells"

_molecules, 2021, doi:10.3390/molecules26072002_

Round 1
Reviewer 1 Report
This present manuscript report the effects of an standardized water extract of KRG (Korean Red Ginseng) and Rg3 (one of its chemical constitutes) on the expression of mucins induced by ASD (Asian Sand Dust).
The manuscript is well written and organized. The authors also provided more insights about the mechanisms involved in ASD action on Human brochial epithelial BEAS-2B cells. This information will be important for other researches. For this reviewer, the paper deserves be published in this journal after minor adjustments. The suggestions are exposed in the attached pdf file.

Author Response
I thank for taking their time to review my article.
I have made some corrections and added some results in the manuscript after going over the reviewer’s comments.
1) Abstract: line 12, the meaning of Rg3 and MUC
Answer) To clarify changed as 'ginsenoside Rg3. Mucin 5AC (MUC5AC)...'
2) Introduction;
Line 29; The organic and inorganic compounds....
Answer) Compounds 'SiO2, Al2O3, Fe2O3, et al.,' was added as 'The organic and inorganic compounds of ASD, such as SiO2, Al2O3, Fe2O3, et al., influence....'
Line 46 - 56, Provide more information about Rg3 ...
Answer) last part of Introduction, 'Ginsenoside Rg3 (Rg3), a member of protopanaxadiols, is the main component of KRG with anti-inflammatory and anti-cancer properties. Anti-inflammatory effects of Rg3 suppress nitric oxide, reactive oxygen species, prostaglandin E2 and proinflammatory cytokine production [18]. Ant-cancer effect is associated with induction of apoptosis, induction of autophagy, and inhibition of angiogenesis[19]. However, the effect of KRG and Rg3 on mucin production in airway epithelial cells are not commonly studied., was added to improve the information of Rg3.
3) Results:
Line 66; add vlaue of IC50
Answer) IC50 for KRG, Rg3, and ASD was added at the last part of 2.1, as'The IC50 values of ASD, KRG, and Rg3 were derived from dose-response curves (24h: KRG: 995 μg/ml , Rg3 210 μg/ml, and ASD 282 μg/ml; 48h: KRG 1130 μg/ml, Rg3 148 μg/ml, and ASD 186 μg/ml; 72h: KRG 613 μg/ml, Rg3 139 μg/ml, and ASD 172 μg/ml respectively).'
2.3 and 2.3: Line 80and 93; added standard deviation
Answer) SD was added as recommended.
Figures: differentiate the graphs using letters
Answer) Figure 1, 2, 3, 4, and 5 was changed as recommend. Text and legend were also changed as recommended.
4) Discussion:
line 138 to 148. To improve the readability and understanding Answer) Sentences were changed as 'Although, not all of the ginsenosides isolated from KRG influenced inflammatory responses, Rg3 has an anti-inflammatory effect via the reduction of COX-2, inducible nitric oxide synthase, and proinflammatory cytokines [22,23]. Ginsenoside Rb1 was found to inhibit lipopolysaccharide-induced MUC5AC expression in human airway epithelial cells [24]. We evaluated the optimal concentration for the inhibition of mucin gene expression by pretreatment of BEAS-2B cells with various concentrations of KRG (50 to 500 μg/ml) and Rg3 (5 to 50 μg/ml). KRG and RG3 inhibited MUC5AC, MUC5B and MUC8 mRNA expression in dose dependent manner (data not shown), with 500 μg/ml of KRG and 50 μg/ml of Rg3 used as an optimal dose in this study.
Other minor problems have been corrected as pointed out.
I hope the revised manuscript will better meet the requirements of the ‘Molecules' for publication.
Thank you.
Reviewer 2 Report
Authors Shin et al. evaluated the effects of Korean Red Ginseng and Rg3 on Asian Sand Dust-Induced MUC5AC, MUC5B, and MUC8 expression in Bronchial Epithelial Cells. Although the concept is interesting and the data presented somewhat suggest that KRG and Rg3 have potential for treating mucus-producing airway inflammatory diseases, the studies conducted are mostly preliminary. Only a single cell line has been used. Further studies using multiple cell lines and mechanistic evaluations are needed. Performing animal studies would complement the in vitro data.
Author Response
I thank for taking their time to review my article.
As pointed out, this study is a preliminary and basic study to find out the effect of Korean Red ginseng on Asian sand dust induced mucus production in bronchial epithelial cells. We are going to study with primary respiratory epithelial cells and supporting cells as well as in vivo study with animal model.
Unfortunately, we could not cover in this manuscript.